# CAUSAL DISCOVERY WITH REINFORCEMENT LEARNING

**Shengyu Zhu**[†] **Ignavier Ng**[§*] **Zhitang Chen**[†]
[†]Huawei Noah's Ark Lab    [§]University of Toronto
[†]{zhushengyu,chenzhitang2}@huawei.com  [§]ignavierng@cs.toronto.edu

## ABSTRACT

Discovering causal structure among a set of variables is a fundamental problem in many empirical sciences. Traditional score-based casual discovery methods rely on various local heuristics to search for a Directed Acyclic Graph (DAG) according to a predefined score function. While these methods, e.g., greedy equivalence search, may have attractive results with infinite samples and certain model assumptions, they are less satisfactory in practice due to finite data and possible violation of assumptions. Motivated by recent advances in neural combinatorial optimization, we propose to use Reinforcement Learning (RL) to search for the DAG with the best scoring. Our encoder-decoder model takes observable data as input and generates graph adjacency matrices that are used to compute rewards. The reward incorporates both the predefined score function and two penalty terms for enforcing acyclicity. In contrast with typical RL applications where the goal is to learn a policy, we use RL as a search strategy and our final output would be the graph, among all graphs generated during training, that achieves the best reward. We conduct experiments on both synthetic and real datasets, and show that the proposed approach not only has an improved search ability but also allows a flexible score function under the acyclicity constraint.

## 1 INTRODUCTION

Discovering and understanding causal mechanisms underlying natural phenomena are important to many disciplines of sciences. An effective approach is to conduct controlled randomized experiments, which however is expensive or even impossible in certain fields such as social sciences (Bollen, 1989) and bioinformatics (Opgen-Rhein and Strimmer, 2007). Causal discovery methods that infer causal relationships from passively observable data are hence attractive and have been an important research topic in the past decades (Pearl, 2009; Spirtes et al., 2000; Peters et al., 2017).

A major class of such causal discovery methods are score-based, which assign a score $\mathcal{S}(\mathcal{G})$, typically computed with the observed data, to each directed graph $\mathcal{G}$ and then search over the space of all Directed Acyclic Graphs (DAGs) for the best scoring:

$$\min_{\mathcal{G}} \ \mathcal{S}(\mathcal{G}), \text{ subject to } \mathcal{G} \in \text{DAGs}. \tag{1}$$

While there have been well-defined score functions such as the Bayesian Information Criterion (BIC) or Minimum Description Length (MDL) score (Schwarz, 1978; Chickering, 2002) and the Bayesian Gaussian equivalent (BGe) score (Geiger and Heckerman, 1994), Problem (1) is generally NP-hard to solve (Chickering, 1996; Chickering et al., 2004), largely due to the combinatorial nature of its acyclicity constraint with the number of DAGs increasing super-exponentially in the number of graph nodes. To tackle this problem, most existing approaches rely on local heuristics to enforce the acyclicity. For example, Greedy Equivalence Search (GES) enforces acyclicity one edge at a time, explicitly checking for the acyclicity constraint when an edge is added. GES is known to find global minimizer with infinite samples under suitable assumptions (Chickering, 2002; Nandy et al., 2018), but this is not guaranteed in the finite sample regime. There are hybrid methods, e.g., the max-min hill climbing method (Tsamardinos et al., 2006), which use constraint-based approaches to reduce

---

[*]Work was done during an internship at Huawei Noah's Ark Lab.

the search space before applying score-based methods. However, this methodology generally lacks a principled way of choosing a problem-specific combination of score functions and search strategies.

Recently, Zheng et al. (2018) introduced a smooth characterization for the acyclicity. With linear models, Problem (1) was then formulated as a continuous optimization problem w.r.t. the weighted graph adjacency matrix by picking a proper loss function, e.g., the least squares loss. Subsequent works Yu et al. (2019) and Lachapelle et al. (2019) have also adopted the evidence lower bound and the negative log-likelihood as loss functions, respectively, and used Neural Networks (NNs) to model the causal relationships. Note that the loss functions in these methods must be carefully chosen in order to apply continuous optimization methods. Unfortunately, many effective score functions, e.g., the generalized score function proposed by Huang et al. (2018) and the independence based score function from Peters et al. (2014), either cannot be represented in closed forms or have very complicated equivalent loss functions, and thus cannot be easily combined with this approach.

We propose to use Reinforcement Learning (RL) to search for the DAG with the best score according to a predefined score function, as outlined in Figure 1. The insight is that an RL agent with stochastic policy can determine automatically where to search given the uncertainty information of the learned policy, which can be updated promptly by the stream of reward signals. To apply RL to causal discovery, we use an encoder-decoder NN model to generate directed graphs from the observed data, which are then used to compute rewards consisting of the predefined score function as well as two penalty terms to enforce acyclicity. We resort to policy gradient and stochastic optimization methods to train the weights of the NNs, and our output is the graph that achieves the best reward, among all graphs generated in the training process. Experiments on both synthetic and real datasets show that our approach has a much improved search ability without sacrificing any flexibility in choosing score functions. In particular, the proposed approach with BIC score outperforms GES with the same score function on Linear Non-Gaussian Acyclic Model (LiNGAM) and linear-Gaussian datasets, and also outperforms recent gradient based methods when the causal relationships are nonlinear.

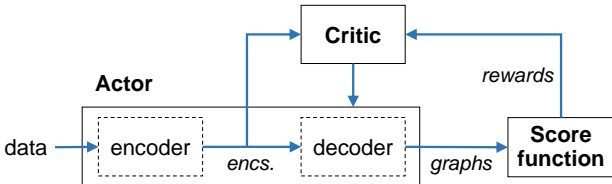

Figure 1: Reinforcement learning for score-based causal discovery.

## 2 RELATED WORK

Constraint-based causal discovery methods first use conditional independence tests to find causal skeleton and then determine the orientations of the edges up to the Markov equivalence class, which usually contains DAGs that can be structurally diverse and may still have many unoriented edges. Examples include Sun et al. (2007); Zhang et al. (2012) that use kernel-based conditional independence criteria and the well-known PC algorithm (Spirtes et al., 2000). This class of methods involve a multiple testing problem where the tests are usually conducted independently. The testing results may have conflicts and handling them is not easy, though there are certain works, e.g., Hyttinen et al. (2014), attempting to tackle this problem. These methods are also not robust as small errors in building the graph skeleton can result in large errors in the inferred Markov equivalence class.

Another class of causal discovery methods are based on properly defined functional causal models. Unlike constraint-based methods that assume faithfulness and identify only the Markov equivalence class, these methods are able to distinguish between different DAGs in the same equivalence class, thanks to the additional assumptions on data distribution and/or functional classes. Examples include LiNGAM (Shimizu et al., 2006; 2011), the nonlinear additive noise model (Hoyer et al., 2009; Peters et al., 2014; 2017), and the post-nonlinear causal model (Zhang and Hyvärinen, 2009).

Besides Yu et al. (2019); Lachapelle et al. (2019), other recent NN based approaches to causal discovery include Goudet et al. (2018) that proposes causal generative NNs to functional causal

modeling with a prior knowledge of initial skeleton of the causal graph and Kalainathan et al. (2018) that learns causal generative models in an adversarial way but does not guarantee acyclicity.

Recent advances in sequence-to-sequence learning (Sutskever et al., 2014) have motivated the use of NNs for optimization in various domains (Vinyals et al., 2015; Zoph and Le, 2017; Chen et al., 2017). A particular example is the traveling salesman problem that was revisited in the work of pointer networks (Vinyals et al., 2015). Authors proposed a recurrent NN with nonparametric softmaxes trained in a supervised manner to predict the sequence of visited cities. Bello et al. (2016) further proposed to use the RL paradigm to tackle the combinatorial problems due to their relatively simple reward mechanisms. It was shown that an RL agent can have a better generalization even when the optimal solutions are used as labeled data in the previous supervised approach. Alternatively, the RL based approach in Dai et al. (2017) considered combinatorial optimization problems on (undirected) graphs and achieved a promising performance by exploiting graph structures, in contrast with the general sequence-to-sequence modeling.

There are many other successful RL applications in recent years, e.g., AlphaGo (Silver et al., 2017), where the goal is to learn a policy for a given task. As an exception, Zoph and Le (2017) applied RL to neural architecture search. While we use a similar idea as the RL paradigm can naturally include the search task, our work is different in the actor and reward designs: our actor is an encoder-decoder model that generates graph adjacency matrices (cf. Section 4) and the reward is tailored for causal discovery by incorporating a score function and the acyclicity constraint (cf. Section 5.1).

## 3  MODEL DEFINITION

We assume the following model for data generating procedure, as in Hoyer et al. (2009); Peters et al. (2014). Each variable $x_i$ is associated with a node $i$ in a $d$-node DAG $\mathcal{G}$, and the observed value of $x_i$ is obtained as a function of its parents in the graph plus an independent additive noise $n_i$, i.e.,

$$x_i := f_i(\mathbf{x}_{\mathrm{pa}(i)}) + n_i, i = 1, 2, \ldots, d,$$

where $\mathbf{x}_{\mathrm{pa}(i)}$ denotes the set of variables $x_j$ so that there is an edge from $x_j$ to $x_i$ in the graph, and the noises $n_i$ are assumed to be jointly independent. We also assume causal minimality, which in this case reduces to that each function $f_i$ is not a constant in any of its arguments (Peters et al., 2014). Without further assumption on the forms of functions and/or noises, the above model can be identified only up to Markov equivalence class under the usual Markov and faithful assumptions (Spirtes et al., 2000; Peters et al., 2014); in our experiments we will consider synthetic datasets that are generated from fully identifiable models so that it is practically meaningful to evaluate the estimated graph w.r.t. the true DAG. If all the functions $f_i$ are linear and the noises $n_i$ are Gaussian distributed, the above model yields the class of standard linear-Gaussian model that has been studied in Bollen (1989); Geiger and Heckerman (1994); Spirtes et al. (2000); Peters et al. (2017). When the functions are linear but the noises are non-Gaussian, one can obtain the LiNGAM described in Shimizu et al. (2006; 2011) and the true DAG can be uniquely identified under favorable conditions.

In this paper, we consider that all the variables $x_i$ are scalars; extending to more complex cases is straightforward, provided with a properly defined score function. The observed data $\mathbf{X}$, consisting of a number of vectors $\mathbf{x} := [x_1, x_2, \ldots, x_d]^T \in \mathbb{R}^d$, are then sampled independently according to the above model on an unknown DAG, with fixed functions $f_i$ and fixed distributions for $n_i$. The objective of causal discovery is to use the observed data $\mathbf{X}$, which gives the empirical version of the joint distribution of $\mathbf{x}$, to infer the underlying causal DAG $\mathcal{G}$.

## 4  NEURAL NETWORK ARCHITECTURE FOR GRAPH GENERATION

Given a dataset $\mathbf{X} = \{\mathbf{x}^k\}_{k=1}^m$ where $\mathbf{x}^k$ denotes the $k$-th observed sample, we want to infer the causal graph that best describes the data generating procedure. We would like to use NNs to infer the causal graph from the observed data; specifically, we aim to design an NN based graph generator whose input is the observed data and the output is a graph adjacency matrix. A naive choice would be using feed-forward NNs to output $d^2$ scalars and then reshape them to an adjacency matrix in $\mathbb{R}^{d \times d}$. However, this NN structure failed to produce promising results, possibly because the feed-forward NNs could not provide sufficient interactions amongst variables to capture the causal relations.

Motivated by recent advances in neural combinatorial optimization, particularly the pointer networks (Bello et al., 2016; Vinyals et al., 2015), we draw $n$ random samples (with replacement) $\{\mathbf{x}^l\}_{l=1}^n$ from $\mathbf{X}$ and reshape them as $\mathbf{s} := \{\tilde{\mathbf{x}}_i\}_{i=1}^d$ where $\tilde{\mathbf{x}}_i \in \mathbb{R}^n$ is the vector concatenating all the $i$-th entries of the vectors in $\{\mathbf{x}^l\}_{l=1}^n$. In an analogy to the traveling salesman problem, this represents a sequence of $d$ cities lying in an $n$-dim space. We are concerned with generating a binary adjacency matrix $A \in \{0,1\}^{d \times d}$ so that the corresponding graph is acyclic and achieves the best score. In this work we consider encoder-decoder models for graph generation:

**Encoder**   We use the attention based encoder in the Transformer structure proposed by Vaswani et al. (2017). We believe that the self-attention scheme, together with structural DAG constraint, is capable of finding the causal relations amongst variables. Other attention based models such as graph attention network (Veličković et al., 2018) may also be used, which will be considered in a future work. Denote the outputs of the encoder by $enc_i, i = 1, 2, \ldots, d$, with dimension $d_e$.

**Decoder**   Our decoder generates the graph adjacency matrix in an element-wise manner, by building relationships between two encoder outputs $enc_i$ and $enc_j$. We consider the single layer decoder

$$g_{ij}(W_1, W_2, u) = u^T \tanh(W_1 \, enc_i + W_2 \, enc_j),$$

where $W_1, W_2 \in \mathbb{R}^{d_h \times d_e}$, $u \in \mathbb{R}^{d_h \times 1}$ are trainable parameters and $d_h$ is the hidden dimension associated with the decoder. To generate a binary adjacency matrix $A$, we pass each entry $g_{ij}$ into a logistic sigmoid function $\sigma(\cdot)$ and then sample according to a Bernoulli distribution with probability $\sigma(g_{ij})$ that indicates the probability of existing an edge from $x_i$ to $x_j$. To avoid self-loops, we simply mask the $(i, i)$-th entry in the adjacency matrix.

Other decoder choices include the neural tensor network model (Socher et al., 2013) and the bilinear model that build the pairwise relationships between encoder outputs. Another choice is the Transformer decoder which can generate an adjacency matrix in a row-wise manner. Empirically, we find that the single layer decoder performs the best, possibly because it contains less parameters and is easier to train to find better DAGs while the self-attention based encoder has provided sufficient interactions amongst the variables for causal discovery. Appendix A provides more details regarding these decoders and their empirical results with linear-Gaussian data models.

## 5   REINFORCEMENT LEARNING FOR SEARCH

In this section, we use RL as our search strategy to find the DAG with the best score, as outlined in Figure 1. As one will see, the proposed method possesses an improved search ability over traditional score-based methods and also allows flexible score functions subject to the acyclicity constraint.

### 5.1   SCORE FUNCTION, ACYCLICITY, AND REWARD

**Score Function**   In this work, we consider only existing score functions to construct the reward that will be maximized by an RL agent. Often score-based methods assume a parametric model for causal relationships (e.g., linear-Gaussian equations or multinomial distribution), which introduces a set of parameters $\theta$. Among all score functions that can be directly included here, we focus on the BIC score that is not only consistent (Haughton et al., 1988) but also locally consistent for its decomposability (Chickering, 1996).

The BIC score for a given directed graph $\mathcal{G}$ is

$$\mathcal{S}_{\text{BIC}}(\mathcal{G}) = -2 \log p(\mathbf{X}; \hat{\theta}, \mathcal{G}) + d_\theta \log m,$$

where $\hat{\theta}$ is the maximum likelihood estimator and $d_\theta$ denotes the dimensionality of the parameter $\theta$. We assume i.i.d. Gaussian additive noises throughout this paper. If we apply linear models to each causal relationship and let $\hat{x}_i^k$ be the corresponding estimate for $x_i^k$, the $i$-th entry in the $k$-th observed sample, then we have the BIC score being (up to some additive constant)

$$\mathcal{S}_{\text{BIC}}(\mathcal{G}) = \sum_{i=1}^d \left( m \log(\text{RSS}_i / m) \right) + \#(\text{edges}) \log m, \tag{2}$$

where $\text{RSS}_i = \sum_{k=1}^m (x_i^k - \hat{x}_i^k)^2$ denotes the residual sum of squares for the $i$-th variable. The first term in Eq. (2) is equivalent to the log-likelihood objective used by GraN-DAG (Lachapelle et al.,

2019) and the second term adds penalty on the number of edges in the graph $\mathcal{G}$. Further assuming that the noise variances are equal (despite the fact that they may be different), we have

$$\mathcal{S}_{\text{BIC}}(\mathcal{G}) = md \log \left( \left( \sum_{i=1}^{d} \text{RSS}_i \right) / (md) \right) + \#(\text{edges}) \log m. \tag{3}$$

We notice that $\sum_i \text{RSS}_i$ is the least squares loss used in NOTEARS (Zheng et al., 2018). Besides assuming linear models, other regression methods can also be used to estimate $x_i^k$. In Section 6, we will use quadratic regression and Gaussian Process Regression (GPR) to model causal relationships based on the observed data.

**Acyclicity**    A remaining issue is the acyclicity constraint. Other than GES that explicitly checks for acyclicity each time an edge is added, we add penalty terms w.r.t. acyclicity to the score function to enforce acyclicity in an implicit manner and allow the generated graph to change more than one edges at each iteration. In this work, we use a recent result from Zheng et al. (2018): a directed graph $\mathcal{G}$ with binary adjacency matrix $A$ is acyclic if and only if

$$h(A) := \text{trace}\left(e^A\right) - d = 0, \tag{4}$$

where $e^A$ is the matrix exponential of $A$. We find that $h(A)$, which is non-negative, can be small for cyclic graphs and the minimum over all non-DAGs is not easy to find. We would require a very large penalty weight to guarantee acyclicity if only $h(A)$ is used. We thus add another penalty term, the indicator function w.r.t. acyclicity, to induce exact DAGs. We remark that other functions (e.g., the total length of all cyclic paths in the graph), which compute some 'distance' from a directed graph to DAGs and need not be smooth, may also be used to construct the acyclicity penalty in our approach.

**Reward**    Our reward incorporates both the score function and the acyclicity constraint:

$$\text{reward} := - \left[ \mathcal{S}(\mathcal{G}) + \lambda_1 \mathbf{I}(\mathcal{G} \notin \text{DAGs}) + \lambda_2 h(A) \right], \tag{5}$$

where $\mathbf{I}(\cdot)$ denotes the indicator function and $\lambda_1, \lambda_2 \geq 0$ are two penalty parameters. It is not hard to see that the larger $\lambda_1$ and $\lambda_2$ are, the more likely a generated graph with a high reward is acyclic. We then aim to maximize the reward over all possible directed graphs, or equivalently, we have

$$\min_{\mathcal{G}} \ \left[ \mathcal{S}(\mathcal{G}) + \lambda_1 \mathbf{I}(\mathcal{G} \notin \text{DAGs}) + \lambda_2 h(A) \right]. \tag{6}$$

An interesting question is whether this new formulation is equivalent to the original problem with hard acyclicity constraint. Fortunately, the following proposition guarantees that Problems (1) and (6) are equivalent with properly chosen $\lambda_1$ and $\lambda_2$, which can be verified by showing that a minimizer of one problem is also a solution to the other. A proof is provided in Appendix B for completeness.

**Proposition 1.** *Let $h_{\min} > 0$ be the minimum of $h(A)$ over all directed cyclic graphs, i.e., $h_{\min} = \min_{\mathcal{G} \notin \text{DAGs}} h(A)$. Let $\mathcal{S}^*$ denote the optimal score achieved by some DAG in Problem (1). Assume that $\mathcal{S}_L \in \mathbb{R}$ is a lower bound of the score function over all possible directed graphs, i.e., $S_L \leq \min_{\mathcal{G}} \mathcal{S}(\mathcal{G})$, and $S_U \in \mathbb{R}$ is an upper bound on the optimal score with $\mathcal{S}^* \leq \mathcal{S}_U$. Then Problems (1) and (6) are equivalent if*

$$\lambda_1 + \lambda_2 h_{\min} \geq \mathcal{S}_U - \mathcal{S}_L.$$

For practical use, we need to find respective quantities in order to choose proper penalty parameters. An upper bound $\mathcal{S}_U$ can be easily found by drawing some random DAGs or using the results from other methods like NOTEARS. A lower bound $\mathcal{S}_L$ depends on the particular score function. With BIC score, we can fit each variable $x_i$ against all the rest variables, and use only the $\text{RSS}_i$ terms but ignore the additive penalty on the number of edges. With the independence based score function proposed by Peters et al. (2014), we may simply set $\mathcal{S}_L = 0$. The minimum term $h_{\min}$, as previously mentioned, may not be easy to find. Fortunately, with $\lambda_1 = \mathcal{S}_U - \mathcal{S}_L$, Proposition 1 guarantees the equivalence of Problems (1) and (6) for any $\lambda_2 \geq 0$. However, simply setting $\lambda_2 = 0$ could only get good performance with very small graphs (see a discussion in Appendix C). We will pick a relatively small value for $\lambda_2$, which helps to generate directed graphs that become closer to DAGs.

Empirically, we find that if the initial penalty weights are set too large, the score function would have little effect on the reward, which then limits the exploration of the RL agent and usually results in DAGs with high scores. Similar to Lagrangian methods, we can start with small penalty weights and gradually increase them so that the condition in Proposition 1 is satisfied. Meanwhile, we notice that

---

**Algorithm 1** The proposed RL approach to score-based causal discovery

---

**Require:** score parameters: $\mathcal{S}_L$, $\mathcal{S}_U$, and $\mathcal{S}_0$; penalty parameters: $\lambda_1$, $\Delta_1$, $\lambda_2$, $\Delta_2$, and $\Lambda_2$; iteration number for parameter update: $t_0$.

1: **for** $t = 1, 2, \ldots$ **do**
2:     Run actor-critic algorithm, with score adjustment by $\mathcal{S} \leftarrow \mathcal{S}_0(\mathcal{S} - \mathcal{S}_L)/(\mathcal{S}_U - \mathcal{S}_L)$
3:     **if** $t \ (\mathrm{mod} \ t_0) = 0$ **then**
4:         **if** the maximum reward corresponds to a DAG with score $\mathcal{S}_{\min}$ **then**
5:             update $\mathcal{S}_U \leftarrow \min(\mathcal{S}_U, \mathcal{S}_{\min})$
6:         **end if**
7:         update $\lambda_1 \leftarrow \min(\lambda_1 + \Delta_1, \mathcal{S}_U)$ and $\lambda_2 \leftarrow \min(\lambda_2\Delta_2, \Lambda_2)$
8:         update recorded rewards according to new $\lambda_1$ and $\lambda_2$
9:     **end if**
10: **end for**

---

different score functions may have different ranges while the acyclicity penalty terms are independent of the particular range of the score function. We hence also adjust the predefined scores to a certain range by using $\mathcal{S}_0(\mathcal{S} - \mathcal{S}_L)/(\mathcal{S}_U - \mathcal{S}_L)$ for some $\mathcal{S}_0 > 0$ and the optimal score will lie in $[0, \mathcal{S}_0]$.[1] Our algorithm is summarized in Algorithm 1, where $\Delta_1$ and $\Delta_2$ are the updating parameters associated with $\lambda_1$ and $\lambda_2$, respectively, and $t_0$ denotes the updating frequency. The weight $\lambda_2$ is updated in a similar manner to the updating rule on the Lagrange multiplier used by NOTEARS and we set $\Lambda_2$ as an upper bound on $\lambda_2$, as previously discussed. In all our experiments that use BIC as score function, $\mathcal{S}_L$ is obtained from a complete directed graph and $\mathcal{S}_U$ is from an empty graph. Since $\mathcal{S}_U$ with the empty graph can be very high for large graphs, we also update it by keeping track of the lowest score achieved by DAGs generated during training. Other parameter choices in this work are $\mathcal{S}_0 = 5$, $t_0 = 1,000$, $\lambda_1 = 0$, $\Delta_1 = 1$, $\lambda_2 = 10^{-\lceil d/3 \rceil}$, $\Delta_2 = 10$ and $\Lambda_2 = 0.01$. We comment that these parameter choices may be further tuned for specific applications, and the inferred causal graph would be the one that is acyclic and achieves the best score, among all the final outputs (cf. Section 5.3) of the RL approach with different parameter choices.

## 5.2 ACTOR-CRITIC ALGORITHM

We believe that the exploitation and exploration scheme in the RL paradigm provides an appropriate way to guide the search. Let $\pi(\cdot \mid \mathbf{s})$ and $\psi$ denote the policy and NN parameters for graph generation, respectively. Our training objective is the expected reward defined as

$$J(\psi \mid \mathbf{s}) = \mathbb{E}_{A \sim \pi(\cdot \mid \mathbf{s})} \left\{ - \left[ \mathcal{S}(\mathcal{G}) + \lambda_1 \mathbf{I}(\mathcal{G} \notin \mathsf{DAGs}) + \lambda_2 h(A) \right] \right\}. \tag{7}$$

During training, the input $\mathbf{s}$ is constructed by randomly drawing samples from the observed dataset $\mathbf{X}$, as described in Section 4.

We resort to policy gradient methods and stochastic methods to optimize the parameters $\psi$. The gradient $\nabla_\psi J(\psi \mid \mathbf{s})$ can be obtained by the well-known REINFORCE algorithm (Williams, 1992; Sutton et al., 2000). We draw $B$ samples $\mathbf{s}_1, \mathbf{s}_2, \ldots, \mathbf{s}_B$ as a batch to estimate the gradient which is then used to train the NNs through stochastic optimization methods like Adam (Kingma and Ba, 2014). Using a parametric baseline to estimate the reward can also help training (Konda and Tsitsiklis, 2000). For the present work, our critic is a simple 2-layer feed-forward NN with ReLU units, with the input being the encoder outputs $\{enc_i\}_{i=1}^d$. The critic is trained with Adam on a mean squared error between its predictions and the true rewards. An entropy regularization term (Williams and Peng, 1991; Mnih et al., 2016) is also added to encourage exploration of the RL agent. Although policy gradient methods only guarantee local convergence under proper conditions (Sutton et al., 2000), we remark that the inferred graphs from the actor-critic algorithm are all DAGs in our experiments.

Training an RL agent typically requires many iterations. In the present work, we find that computing the rewards for generated graphs is much more time-consuming than training NNs. Therefore, we record the computed rewards corresponding to different graph structures. Moreover, the BIC score can be decomposed according to single causal relationships and we also record the corresponding $\mathrm{RSS}_i$ to avoid repeated computations.

---

[1]When $\mathcal{S}_U - \mathcal{S}_L = 0$, then we have obtained the solution if we know the graph that achieves $\mathcal{S}_U$ or $\mathcal{S}_L$, or otherwise we may simply pick a new upper bound as $\mathcal{S}_U + 1$.

### 5.3 FINAL OUTPUT

Since we are concerned with finding a DAG with the best score rather than a policy, we record all the graphs generated during the training process and output the one with the best reward. In practice, the graph may contain spurious edges and further processing is needed.

To this end, we can prune the estimated edges in a greedy way, according to either the regression performance or the score function. For an inferred causal relationship, we remove a parental variable and calculate the performance of the resulting graph, with all other causal relationships unchanged. If the performance does not degrade or degrade within a predefined tolerance, we accept pruning and continue this process with the pruned causal relationship. For linear models, pruning can be simply done by thresholding the estimated coefficients.

Related to the above pruning process is to add to the score function an increased penalty weight on the number of edges of a graph. However, this weight is not easy to choose, as a large weight may incur missing edges. In this work, we stick to the penalty weight $\log m$ that is included in the BIC score and then apply pruning to the inferred graph in order to reduce false discoveries.

## 6 EXPERIMENTAL RESULTS

We report empirical results on synthetic and real datasets to compare our approach against both traditional and recent gradient based approaches, including GES (with BIC score) (Chickering, 2002; Ramsey et al., 2017), the PC algorithm (with Fisher-z test and $p$-value 0.01) (Spirtes et al., 2000), ICA-LiNGAM (Shimizu et al., 2006), the Causal Additive Model (CAM) based algorithm proposed by Bühlmann et al. (2014), NOTEARS (Zheng et al., 2018), DAG-GNN (Yu et al., 2019), and GraN-DAG (Lachapelle et al., 2019), among others. All these algorithms have available implementations and we give a brief description on these algorithms and their implementations in Appendix D. Default hyper-parameters of these implementations are used unless otherwise stated. For pruning, we use the same thresholding method for ICA-LiNGAM, NOTEARS, and DAG-GNN. Since the authors of CAM and GraN-DAG propose to apply significance testing of covariates based on generalized additive models and then declare significance if the reported $p$-values are lower than or equal to 0.001, we stick to the same pruning method for CAM and GraN-DAG.

The proposed RL based approach is implemented based on an existing Tensorflow (Abadi et al., 2016) implementation of neural combinatorial optimizer (see Appendix D for more details). The decoder is modified as described in Section 4 and the RL algorithm related hyper-parameters are left unchanged. We pick $B = 64$ as batch size at each iteration and $d_h = 16$ as the hidden dimension with the single layer decoder. Our approach is combined with the BIC scores under Gaussianity assumption given in Eqs. (2) and (3), and are denoted as RL-BIC and RL-BIC2, respectively.

We evaluate the estimated graphs using three metrics: False Discovery Rate (FDR), True Positive Rate (TPR), and Structural Hamming Distance (SHD) which is the smallest number of edge additions, deletions, and reversals to convert the estimated graph into the true DAG. The SHD takes into account both false positives and false negatives and a lower SHD indicates a better estimate of the causal graph. Since GES and PC may output unoriented edges, we follow Zheng et al. (2018) to treat GES and PC favorably by regarding undirected edges as true positives as long as the true graph has a directed edge in place of the undirected edge.

### 6.1 LINEAR MODEL WITH GAUSSIAN AND NON-GAUSSIAN NOISE

Given number of variables $d$, we generate a $d \times d$ upper triangular matrix as the graph binary adjacency matrix, in which the upper entries are sampled independently from $\text{Bern}(0.5)$. We assign edge weights independently from $\text{Unif}([-2, -0.5] \cup [0.5, 2])$ to obtain a weight matrix $W \in \mathbb{R}^{d \times d}$, and then sample $\mathbf{x} = W^T \mathbf{x} + \mathbf{n} \in \mathbb{R}^d$ from both Gaussian and non-Gaussian noise models. The non-Gaussian noise is the same as the one used for ICA-LiNGAM (Shimizu et al., 2006), which generates samples from a Gaussian distribution and passes them through a power nonlinearity to make them non-Gaussian. We pick unit noise variances in both models and generate $m = 5,000$ samples as our datasets. A random permutation of variables is then performed. This data generating procedure is similar to that used by NOTEARS and DAG-GNN and the true causal graphs in both cases are known to be identifiable (Shimizu et al., 2006; Peters and Bühlmann, 2013).

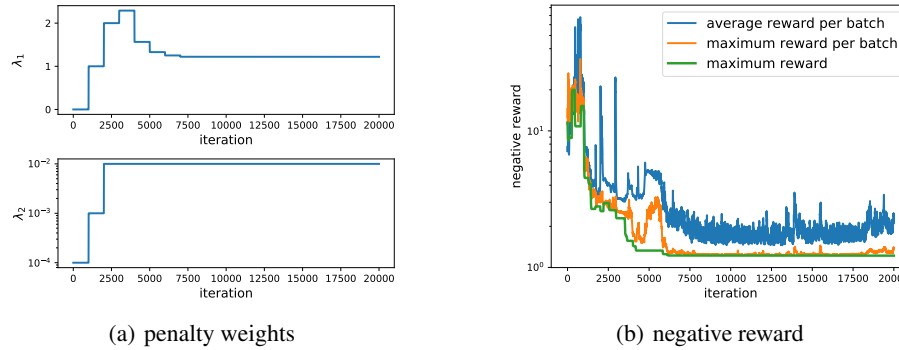

(a) penalty weights  (b) negative reward

Figure 2: Learning process of the proposed method RL-BIC2 on a linear-Gaussian dataset.

Table 1: Empirical results on LiNGAM and linear-Gaussian data models with 12-node graphs.

|  |  | RL-BIC | RL-BIC2 | PC | GES | ICA-LiNGAM | CAM | NOTEARS | DAG-GNN | GraN-DAG |
|---|---|---|---|---|---|---|---|---|---|---|
| LiNGAM | FDR | $0.28 \pm 0.11$ | $0 \pm 0$ | $0.06 \pm 0.04$ | $0.62 \pm 0.06$ | $0 \pm 0$ | $0.67 \pm 0.08$ | $0.04 \pm 0.03$ | $0.11 \pm 0.03$ | $0.63 \pm 0.10$ |
|  | TPR | $0.71 \pm 0.17$ | $1 \pm 0$ | $0.25 \pm 0.03$ | $0.25 \pm 0.04$ | $1 \pm 0$ | $0.49 \pm 0.07$ | $0.95 \pm 0.05$ | $0.94 \pm 0.04$ | $0.37 \pm 0.15$ |
|  | SHD | $17.4 \pm 7.50$ | $0 \pm 0$ | $31.8 \pm 2.04$ | $32.8 \pm 2.93$ | $0 \pm 0$ | $40.4 \pm 5.92$ | $2.4 \pm 2.42$ | $5.00 \pm 1.41$ | $36.0 \pm 5.33$ |
| Linear-Gaussian | FDR | $0.38 \pm 0.13$ | $0 \pm 0$ | $0.52 \pm 0.07$ | $0.63 \pm 0.06$ | $0.65 \pm 0.02$ | $0.70 \pm 0.08$ | $0.02 \pm 0.02$ | $0.10 \pm 0.05$ | $0.70 \pm 0.17$ |
|  | TPR | $0.66 \pm 0.12$ | $1 \pm 0$ | $0.31 \pm 0.03$ | $0.24 \pm 0.04$ | $0.73 \pm 0.05$ | $0.44 \pm 0.11$ | $0.98 \pm 0.02$ | $0.95 \pm 0.05$ | $0.27 \pm 0.13$ |
|  | SHD | $22.2 \pm 6.34$ | $0 \pm 0$ | $29.6 \pm 3.01$ | $33.2 \pm 2.48$ | $46.2 \pm 2.79$ | $40.8 \pm 4.53$ | $1.0 \pm 0.89$ | $4.40 \pm 2.06$ | $38.2 \pm 6.68$ |

We first consider graphs with $d = 12$ nodes. We use $n = 64$ for constructing the input sample and set the maximum number of iterations to $20,000$. We use a threshold $0.3$, same as NOTEARS and DAG-GNN with this data model, to prune the estimated edges. Figure 2 shows the learning process of the proposed method RL-BIC2 on a linear-Gaussian dataset. In this example, RL-BIC2 generates $683,784$ different graphs during training, much lower than the total number (around $5.22 \times 10^{26}$) of DAGs. The pruned DAG turns out to be exactly the same as the underlying causal graph.

We report the empirical results on LiNGAM and linear-Gaussian data models in Table 1. Both PC and GES perform poorly, possibly because we consider relatively dense graphs for our data generating procedure. CAM does not perform well either, as it assumes nonlinear causal relationships. ICA-LiNGAM recovers all the true causal graphs for LiNGAM data but performs poorly on linear-Gaussian data. This is not surprising because ICA-LiNGAM works for non-Gaussian noise and does not provide guarantee for linear-Gaussian datasets. Both NOTEARS and DAG-GNN have good causal discovery results whereas GraN-DAG performs much worse. We believe that it is because GraN-DAG uses 2-layer feed-forward NNs to model the causal relationships, which may not be able to learn a good linear relationship in this experiment. Modifying the feed-forward NNs to linear functions reduces to NOTEARS with negative log-likelihood as loss function, which yields similar performance on these datasets (see Appendix E.1 for detailed results). As to our proposed methods, we observe that RL-BIC2 recovers all the true causal graphs on both data models in this experiment while RL-BIC has a worse performance. One may wonder whether this observation is due to the same noise variances that are used in our data models; we conduct additional experiments where the noise variances are randomly sampled and RL-BIC2 still outperforms RL-BIC by a large margin (see also Appendix E.1). Nevertheless, with the same BIC score, RL-BIC performs much better than GES on both datasets, indicating that the RL approach brings in a greatly improved search ability.

Finally, we test the proposed method on larger graphs with $d = 30$ nodes, where the upper entries are sampled independently from $\mathrm{Bern}(0.2)$. This edge probability choice corresponds to the fact that large graphs usually have low edge degrees in practice; see, e.g., the experiment settings of Zheng et al. (2018); Yu et al. (2019); Lachapelle et al. (2019). To incorporate this prior information in our approach, we add to each $g_{ij}$ a common bias term initialized to $-10$ (see Appendix E.1 for details). Considering the much increased search space, we also choose a larger number of observed samples, $n = 128$, to construct the input for graph generator and increase the training iterations to $40,000$. On LiNGAM datasets, RL-BIC2 has FDR, TPR, and SHD being $0.14 \pm 0.15$, $0.94 \pm 0.07$, and $19.8 \pm 23.0$, respectively, comparable to NOTEARS with $0.13 \pm 0.09$, $0.94 \pm 0.04$, and $17.2 \pm 13.12$.

## 6.2 Nonlinear Model with Quadratic Functions

We now consider nonlinear causal relationships with quadratic functions. We generate an upper triangular matrix in a similar way to the first experiment. For a causal relationship with parents $\mathbf{x}_{\mathrm{pa}(i)} = [x_{i_1}, x_{i_2}, \ldots]^T$ at the $i$-th node, we expand $\mathbf{x}_{\mathrm{pa}(i)}$ to contain both first- and second-order features. The coefficient for each term is then either 0 or sampled from Unif $([-1, -0.5] \cup [0.5, 1])$, with equal probability. If a parent variable does not appear in any feature term with a non-zero coefficient, then we remove the corresponding edge in the causal graph. The rest follows the same as in first experiment and here we use the non-Gaussian noise model with 10-node graphs and $5,000$ samples. The true causal graph is identifiable according to Peters et al. (2014). For this quadratic model, there may exist very large variable values which cause computation issues for quadratic regression. We treat these samples as outliers and detailed processing is given in Appendix E.2.

We use quadratic regression for a given causal relationship and calculate the BIC score (assuming equal noise variances) in Eq. (3). For pruning, we simply apply thresholding, with threshold as 0.3, to the estimated coefficients of both first- and second-order terms. If the coefficient of a second-order term, e.g., $x_{i_1} x_{i_2}$, is non-zero after thresholding, then we have two directed edges that are from $x_{i_1}$ to $x_i$ and from $x_{i_2}$ to $x_i$, respectively. We do not consider PC and GES in this experiment due to their poor performance in the first experiment. Our results with 10-node graphs are reported in Table 2, which shows that RL-BIC2 achieves the best performance.

Table 2: Empirical results on nonlinear models with quadratic functions.

|  | RL-BIC2 | NOTEARS | NOTEARS-2 | NOTEARS-3 | ICA-LiNGAM | CAM | DAG-GNN | GraN-DAG |
|---|---|---|---|---|---|---|---|---|
| FDR | $0.02 \pm 0.04$ | $0.35 \pm 0.06$ | $0.15 \pm 0.10$ | $0 \pm 0$ | $0.47 \pm 0.06$ | $0.32 \pm 0.17$ | $0.39 \pm 0.04$ | $0.40 \pm 0.17$ |
| TPR | $0.98 \pm 0.04$ | $0.71 \pm 0.16$ | $0.70 \pm 0.15$ | $0.79 \pm 0.20$ | $0.76 \pm 0.09$ | $0.78 \pm 0.05$ | $0.55 \pm 0.14$ | $0.73 \pm 0.16$ |
| SHD | $0.6 \pm 1.20$ | $14.8 \pm 3.37$ | $8.8 \pm 3.82$ | $5.2 \pm 5.19$ | $20.4 \pm 5.00$ | $14.1 \pm 5.12$ | $18.0 \pm 2.45$ | $39.6 \pm 5.85$ |

For fair comparison, we apply the same quadratic regression based pruning method to the outputs of NOTEARS, denoted as NOTEARS-2. We see that this pruning further reduces FDR, i.e., removes spurious edges, with little effect on TPR. Since pruning does not help discover additional positive edges or increase TPR, we will not apply this pruning method to other methods as their TPRs are much lower than that of RL-BIC2. Finally, with prior knowledge that the function form is quadratic, we can modify NOTEARS to apply quadratic functions to modeling the causal relationships, with an equivalent weighted adjacency matrix constructed using the coefficients of the first- and second-order terms, similar to the idea used by GraN-DAG (detailed derivations are given in Appendix E.2). The problem then becomes a nonconvex optimization problem with $(d-1)d^2/2$ parameters (which are the coefficients of both first- and second-order features), compared to the original NOTEARS with $d^2$ parameters. This method corresponds to NOTEARS-3 in Table 2. Despite the fact that NOTEARS-3 did not achieve a better overall performance than RL-BIC2, we comment that it discovered almost correct causal graphs (with SHD $\leq 2$) on more than half of the datasets, but performed poorly on the rest datasets. We believe that it is due to the increased number of optimization parameters and the more complicated equivalent adjacency matrix which make the optimization problem harder to solve. Meanwhile, we do not exclude that NOTEARS-3 can achieve a better causal discovery performance with other optimization algorithms.

## 6.3 Nonlinear Model with Gaussian Processes

Given a randomly generated causal graph, we consider another nonlinear model where each causal relationship $f_i$ is a function sampled from a Gaussian process with RBF kernel of bandwidth one. The additive noise $n_i$ is normally distributed with variance sampled uniformly. This setting is known to be identifiable according to Peters et al. (2014). We use a setup that is also considered by GraN-DAG (Lachapelle et al., 2019): 10-node and 40-edge graphs with $1,000$ generated samples.

The empirical results are reported in Table 3. One can see that ICA-LiNGAM, NOTEARS, and DAG-GNN perform poorly on this data model. A possible reason is that they may not be able to model this type of causal relationship. More importantly, these methods operate on a notion of weighted adjacency matrix, which is not obvious here. For our method, we apply Gaussian Process Regression (GPR) with RBF kernel to model the causal relationships. Notice that even though the observed data are from a function sampled from Gaussian process, it is not guaranteed that GPR

with the same kernel can achieve a good performance. Indeed, using a fixed kernel bandwidth would lead to severe overfitting that incurs many spurious edges and the graph with the highest reward is usually not a DAG. To proceed, we normalize the observed data and apply median heuristics for kernel bandwidth. Both our methods perform reasonably well, with RL-BIC outperforming all the other methods.

Table 3: Empirical results on nonlinear models with Gaussian processes.

|  | RL-BIC | RL-BIC2 | ICA-LiNGAM | NOTEARS | DAG-GNN | GraN-DAG | CAM |
|---|---|---|---|---|---|---|---|
| FDR | $0.14 \pm 0.03$ | $0.17 \pm 0.12$ | $0.48 \pm 0.04$ | $0.48 \pm 0.19$ | $0.36 \pm 0.11$ | $0.12 \pm 0.08$ | $0.15 \pm 0.07$ |
| TPR | $0.96 \pm 0.03$ | $0.80 \pm 0.09$ | $0.63 \pm 0.07$ | $0.18 \pm 0.09$ | $0.07 \pm 0.03$ | $0.81 \pm 0.05$ | $0.82 \pm 0.04$ |
| SHD | $6.2 \pm 1.33$ | $12.0 \pm 5.18$ | $30.4 \pm 2.50$ | $33.8 \pm 2.56$ | $34.6 \pm 1.36$ | $10.2 \pm 2.39$ | $10.2 \pm 2.93$ |

## 6.4 REAL DATA

We consider a real dataset to discover a protein signaling network based on expression levels of proteins and phospholipids (Sachs et al., 2005). This dataset is a common benchmark in graphical models, with experimental annotations well accepted by the biological community. Both observational and interventional data are contained in this dataset. Since we are interested in using observational data to infer causal mechanisms, we only consider the observational data with $m = 853$ samples. The ground truth causal graph given by Sachs et al. (2005) has 11 nodes and 17 edges.

Notice that the true graph is indeed sparse and an empty graph can have an SHD as low as 17. Therefore, we report more detailed results regarding the estimated graph: number of total edges, number of correct edges, and the SHD. Both PC and GES output too many unoriented edges, and we will not report their results here. We apply GPR with RBF kernel to modeling the causal relationships, with the same data normalization and median heuristics for kernel bandwidth as in Section 6.3. We also use CAM pruning on the inferred graph from the training process. The empirical results are given in Table 4. Both RL-BIC and RL-BIC2 achieve promising results, compared with other methods.

Table 4: Empirical results on Sachs dataset.

|  | RL-BIC | RL-BIC2 | ICA-LiNGAM | CAM | NOTEARS | DAG-GNN | GraN-DAG |
|---|---|---|---|---|---|---|---|
| Total Edges | 10 | 10 | 8 | 10 | 20 | 15 | 10 |
| Correct Edges | 6 | 7 | 4 | 6 | 6 | 6 | 5 |
| SHD | 12 | 11 | 14 | 12 | 19 | 16 | 13 |

## 7 CONCLUDING REMARKS AND FUTURE WORKS

We have proposed to use RL to search for the DAG with the optimal score. Our reward is designed to incorporate a predefined score function and two penalty terms to enforce acyclicity. We use the actor-critic algorithm as our RL algorithm, where the actor is constructed based on recently developed encoder-decoder models. Experiments are conducted on both synthetic and real datasets to show the advantages of our method over other causal discovery methods.

We have also shown the effectiveness of the proposed method with 30-node graphs, yet dealing with large graphs (with more than 50 nodes) is still challenging. Nevertheless, many real applications, like Sachs dataset (Sachs et al., 2005), have a relatively small number of variables. Furthermore, it is possible to decompose large causal discovery problems into smaller ones; see, e.g., Ma et al. (2008). Prior knowledge or constraint-based methods is also applicable to reduce the search space.

There are several future directions from the present work. In our current implementation, computing scores is much more time consuming than training NNs. We believe that developing a more efficient and effective score function will further improve the proposed approach. Other powerful RL algorithms may also be used. For example, the asynchronous advantage actor-critic algorithm has been shown to be effective in many applications (Mnih et al., 2016; Zoph and Le, 2017). In addition, we observe that the total iteration numbers used in our experiments are usually more than needed (see, e.g., Figure 2(b)). A proper early stopping criterion will be favored.

ACKNOWLEDGMENTS

The authors are grateful to the anonymous reviewers for valuable comments and suggestions. The authors would also like to thank Prof. Jiji Zhang from Lingnan University, Dr. Yue Liu from Huawei Noah's Ark Lab, and Zhuangyan Fang from Peking University for many helpful discussions.

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

APPENDIX

## A MORE DETAILS ABOUT DECODERS

We briefly describe the NN based decoders for generating binary adjacency matrices:

- Single layer decoder:

$$g_{ij}(W_1, W_2, u) = u^T \tanh(W_1 \, enc_i + W_2 \, enc_j),$$

  where $W_1, W_2 \in \mathbb{R}^{d_h \times d_e}$, $u \in \mathbb{R}^{d_h \times 1}$ are trainable parameters and $d_h$ denotes the hidden dimension associated with the decoder.
- Bilinear decoder:

$$g_{ij}(W) = enc_i^T W enc_j,$$

  where $W \in \mathbb{R}^{d_e \times d_e}$ is a trainable parameter.
- Neural Tensor Network (NTN) decoder (Socher et al., 2013):

$$g_{ij}(W^{[1:K]}, V, b) = u^T \tanh\left(enc_i^T W^{[1:K]} enc_j + V[enc_i^T, enc_j^T]^T + b\right),$$

  where $W^{[1:k]} \in \mathbb{R}^{d_e \times d_e \times K}$ is a tensor and the bilinear tensor product $enc_i^T W^{[1:K]} enc_j$ results in a vector with each entry being $enc_i^T W^{[k]} enc_j$ for $k = 1, 2, \ldots, K$, $V \in \mathbb{R}^{K \times 2d_e}$, $u \in \mathbb{R}^{K \times 1}$, and $b \in \mathbb{R}^{K \times 1}$.
- Transformer decoder uses the multi-head attention module to obtain the decoder outputs $\{dec_i\}$, followed by a feed-forward NN whose weights are shared across all $dec_i$ (Vaswani et al., 2017). The output consists of $d$ vectors in $\mathbb{R}^d$ which are treated as the row vectors of a $d \times d$ matrix. We then pass each element of this matrix into sigmoid functions and sample a binary adjacency matrix accordingly.

Table 5 provides the empirical results on linear-Gaussian data models with 12-node graphs and unit variances (see Section 6.1 for more details on this data generating procedure). In our implementation, we pick $d_e = 64$ as the dimension of the encoder output, $d_h = 16$ for the single layer decoder, and $K = 2$ for the NTN decoder. We find that single layer decoder performs the best, possibly because it has less parameters and is easier to train to find better DAGs, while the Transformer encoder has provided sufficient interactions amongst variables.

Table 5: Empirical results of different decoders on linear-Gaussian data models with 12-node graphs.

|  |  | Single layer | Bilinear | NTN | Transformer |
|---|---|---|---|---|---|
| Linear-Gaussian | FDR | $0 \pm 0$ | $0.07 \pm 0.13$ | $0.12 \pm 0.14$ | $0.67 \pm 0.07$ |
|  | TPR | $1 \pm 0$ | $0.95 \pm 0.08$ | $0.90 \pm 0.11$ | $0.32 \pm 0.09$ |
|  | SHD | $0 \pm 0$ | $4.0 \pm 7.04$ | $7.4 \pm 8.52$ | $38.2 \pm 3.12$ |

## B EQUIVALENCE OF PROBLEMS (1) AND (6)

*Proof of Proposition 1.* Let $\mathcal{G}$ be a solution to Problem (1). Then we have $\mathcal{S}^* = \mathcal{S}(\mathcal{G})$ and $\mathcal{G}$ must be a DAG due to the hard acyclicity constraint. Assume that $\mathcal{G}$ is not a solution to Problem (6), which indicates that there exists a directed graph $\mathcal{G}'$ (with binary adjacency matrix $A'$) so that

$$\mathcal{S}^* > \mathcal{S}(\mathcal{G}') + \lambda_1 \mathbf{I}(\mathcal{G}' \notin \mathsf{DAGs}) + \lambda_2 h(A'). \tag{8}$$

Clearly, $\mathcal{G}'$ cannot be a DAG, for otherwise we would have a DAG that achieves a lower score than the minimum $\mathcal{S}^*$. By our assumption, it follows that

$$\text{r.h.s. of Eq. (8)} \geq \mathcal{S}_L + \lambda_1 + \lambda_2 h_{\min} \geq \mathcal{S}_U,$$

which contradicts the fact that $\mathcal{S}_U$ is an upper bound on $\mathcal{S}^*$.

For the other direction, let $\mathcal{G}$ be a solution to Problem (6) but not a solution to Problem (1). This indicates that either $\mathcal{G}$ is not a DAG or $\mathcal{G}$ is a DAG but has a higher score than the minimum score, i.e., $\mathcal{S}(\mathcal{G}) > \mathcal{S}^*$. The latter case clearly contradicts the definition of the minimum score. For the former case, assume that some DAG $\mathcal{G}'$ achieves the minimum score. Then plugging $\mathcal{G}'$ into the negative reward, we can get the same inequality in Eq. (8) since both penalty terms are zeros for a DAG. This then contradicts the assumption that $\mathcal{G}$ minimizes the negative reward. □

## C    PENALTY WEIGHT CHOICE

Although setting $\lambda_2 = 0$, or equivalently using only the indicator function w.r.t. acyclicity, can still make Problem (6) equivalent to the original problem with hard acyclicity constraint, we remark that this choice usually does not result in good performance of the RL approach, largely due to that the reward with only the indicator term is likely to fail to guide the RL agent to generate DAGs.

To see why it is the case, consider two cyclic directed graphs, one with all the possible directed edges in place and the other with only two edges (i.e., $x_i \to x_j$ and $x_j \to x_i$ for some $i \neq j$). The latter is much 'closer' to acyclicity in many senses, such as $h(A)$ given in Eq. (4) and number of edge deletion, addition, and reversal to make a directed graph acyclic. Assume a linear data model that has a relatively dense graph. Then the former graph will have a lower BIC score when using linear regression for fitting causal relations, yet the penalty terms of acyclicity are the same with only the indicator function. The former graph then has a higher reward, which does not help the agent to tend to generate DAGs. Also notice that the graphs in our approach are generated according to Bernoulli distributions determined by NN parameters that are randomly initialized. Without loss of generality, consider that each edge is drawn independently according to $\mathrm{Bern}(0.5)$. For small graphs (with less than or equal to 6 nodes or so), a few hundreds of samples of directed graphs are very likely to contain a DAG. Yet for large graphs, the probability of sampling a DAG is much lower. If no DAG is generated during training, the RL agent can hardly learn to generate DAGs. The above facts indeed motivate us to choose a small value of $\lambda_2$ so that the agent can be trained to produce graphs closer to acyclicity and finally to generate exact DAGs.

A question is then what if the RL approach starts with a DAG, e.g., by initializing the probability of generating each edge to be nearly zero. This setting did not lead to good performance, either. The generated directed graphs at early iterations can be very different from the true graphs in that many true edges do not exist, and the resulting score is much higher than the minimum under the DAG constraint. With small penalty weights of the acyclicity terms, the agent could be trained to produce cyclic graphs with better scores, similar to the case with randomly initialized NN parameters. On the other hand, large initial penalty weights, as we have discussed in the paper, limit exploration of the RL agent and usually result in DAGs whose scores are far from optimum.

## D    IMPLEMENTATION DETAILS

We use existing implementations of causal discovery algorithms in comparison, listed below:

- ICA-LiNGAM (Shimizu et al., 2006) assumes linear non-Gaussian additive model for data generating procedure and applies Independent Component Analysis (ICA) to recover the weighted adjacency matrix, followed by thresholding on the weights before outputting the inferred graph. A Python implementation is available at the first author's website `https://sites.google.com/site/sshimizu06/lingam`.

- GES and PC: we use the fast greedy search implementation of GES (Ramsey et al., 2017) which has been reported to outperform other techniques such as max-min hill climbing (Han et al., 2016; Zheng et al., 2018). Implementations of both methods are available through the `py-causal` package at `https://github.com/bd2kccd/py-causal`, written in highly optimized Java codes.

- CAM (Peters et al., 2014) decouples the causal order search among the variables from feature or edge selection in a DAG. CAM also assumes additive noise as in our work, with an additional condition that each function is nonlinear. Codes are available through the CRAN R package repository at `https://cran.r-project.org/web/packages/CAM`.

- NOTEARS (Zheng et al., 2018) recovers the causal graph by estimating the weighted adjacency matrix with the least squares loss and the smooth characterization for acyclicity constraint, followed by thresholding on the estimated weights. Codes are available at the first author's github repository `https://github.com/xunzheng/notears`. We also re-implement the augmented Lagrangian method following the same updating rule on the Lagrange multiplier and the penalty parameter in Tensorflow, so that the augmented Lagrangian at each iteration can be readily minimized without the need of obtaining closed-form gradients. We use this implementation in Sections 6.2 and 6.3 when the objective function and/or the acyclicity constraint are modified.

- DAG-GNN (Yu et al., 2019) formulates causal discovery in the framework of variational autoencoder, where the encoder and decoder are two shallow graph NNs. With a modified smooth characterization on acyclicity, DAG-GNN optimizes a weighted adjacency matrix with the evidence lower bound as loss function. Python codes are available at the first author's github repository `https://github.com/fishmoon1234/DAG-GNN`.

- GraN-DAG (Lachapelle et al., 2019) uses feed-foward NNs to model each causal relationship and chooses the sum of all product paths between variables $x_i$ and $x_j$ as the $(i, j)$-th element of an equivalent weighted adjacency matrix. GraN-DAG uses the same smooth constraint from Zheng et al. (2018) to find a DAG that maximizes the log-likelihood of the observed samples. Codes are available at the first author's github repository `https://github.com/kurowasan/GraN-DAG`.

Our implementation is based on an existing Tensorflow implementation of neural combinatorial optimizer that is available at `https://github.com/MichelDeudon/neural-combinatorial-optimization-rl-tensorflow`. We add an entropy regularization term, and modify the reward and decoder as described in Sections 4 and 5.1, respectively.

## E  Further Experiment Details and Results

### E.1  Experiment 1 in Section 6.1

We replace the feed-forward NNs with linear functions in GraN-DAG and obtain similar experimental results as NOTEARS (FDR, TPR, SHD): $0.05 \pm 0.04$, $0.93 \pm 0.06$, $3.2 \pm 2.93$ and $0.05 \pm 0.04$, $0.95 \pm 0.03$, $2.40 \pm 1.85$ for LiNGAM and linear-Gaussian data models, respectively.

We conduct additional experiments with linear models where the noise variances are uniformly sampled according to $\text{Unif}([0.5, 2])$. Results are given in Table 6.

Table 6: Empirical results on LiNGAM and linear-Gaussian data models with 12-node graphs and different noise variances.

|  |  | RL-BIC | RL-BIC2 | PC | GES | ICA-LiNGAM | CAM | NOTEARS | DAG-GNN | GraN-DAG |
|---|---|---|---|---|---|---|---|---|---|---|
| LiNGAM | FDR | $0.29 \pm 0.12$ | $0.09 \pm 0.06$ | $0.57 \pm 0.10$ | $0.59 \pm 0.13$ | $0 \pm 0$ | $0.70 \pm 0.07$ | $0.08 \pm 0.10$ | $0.14 \pm 0.07$ | $0.71 \pm 0.10$ |
|  | TPR | $0.77 \pm 0.15$ | $0.94 \pm 0.03$ | $0.28 \pm 0.06$ | $0.27 \pm 0.10$ | $1 \pm 0$ | $0.45 \pm 0.12$ | $0.94 \pm 0.07$ | $0.91 \pm 0.04$ | $0.25 \pm 0.09$ |
|  | SHD | $14.4 \pm 7.17$ | $4.0 \pm 2.61$ | $30.4 \pm 4.13$ | $32.0 \pm 5.18$ | $0 \pm 0$ | $41.6 \pm 3.32$ | $3.20 \pm 3.97$ | $6.6 \pm 1.02$ | $38.7 \pm 4.86$ |
| Linear-Gaussian | FDR | $0.36 \pm 0.07$ | $0.10 \pm 0.07$ | $0.54 \pm 0.10$ | $0.61 \pm 0.14$ | $0.67 \pm 0.05$ | $0.65 \pm 0.10$ | $0.07 \pm 0.09$ | $0.12 \pm 0.04$ | $0.71 \pm 0.12$ |
|  | TPR | $0.68 \pm 0.09$ | $0.93 \pm 0.04$ | $0.29 \pm 0.05$ | $0.26 \pm 0.11$ | $0.75 \pm 0.06$ | $0.51 \pm 0.14$ | $0.95 \pm 0.06$ | $0.94 \pm 0.04$ | $0.21 \pm 0.08$ |
|  | SHD | $18.8 \pm 3.43$ | $4.6 \pm 3.07$ | $30.0 \pm 2.83$ | $32.2 \pm 5.42$ | $49.0 \pm 4.82$ | $37.8 \pm 6.31$ | $3.0 \pm 3.58$ | $5.4 \pm 2.06$ | $39.6 \pm 5.85$ |

Knowing a sparse true causal graph *a priori* is also helpful. To incorporate this information in our experiment with 30-node graphs, we add an additional biased term $\tilde{c} \in \mathbb{R}$ to each decoder output: for the single layer decoder, we have

$$g_{ij}(W_1, W_2, u) = u^T \tanh(W_1 \, enc_i + W_2 \, enc_j) + \tilde{c},$$

where we let $\tilde{c}$ be trainable and other parameters have been defined in Appendix A. In our experiments, $\tilde{c}$ is initialized to be $-10$; this choice aims to set a good starting point for generating graph adjacency matrices, motivated by the fact that a good starting point is usually helpful to locally convergent algorithms.

### E.2 EXPERIMENT 2 IN SECTION 6.2

To remove 'outlier' samples with large values that may cause computation issues for quadratic regression, we sort the samples in ascending order according to their $\ell_2$-norms and then pick the first $3,000$ samples for causal discovery.

We use a similar idea from GraN-DAG to build an equivalent weighted adjacency matrix for NOTEARS. Take the first variable $x_1$ for example. We first expand the rest variables $x_2, \ldots, x_d$ to contain both first- and second-order features: $x_2, \ldots, x_d, x_2 x_3, \ldots, x_i x_j, \ldots, x_{d-1} x_d$ with $i, j = 2, \ldots, d$ and $i \leq j$. There are in total $d(d-1)/2$ terms and we use $\mathbf{x}_1$ to denote the vector that concatenates these feature terms. Correspondingly, we use $c_i$ and $c_{ij}$ to denote the coefficients associated with these features and $\mathbf{c}_1$ to denote the concatenating vector of the coefficients. Notice that the variable $x_l$, $l \neq 1$ affects $x_1$ only through the terms $x_l$, $x_i x_l$ with $i \leq l$, and $x_l x_j$ with $j > l$. Therefore, an equivalent weighted adjacency matrix $W$ lying in $\mathbb{R}^{d \times d}$ can be constructed with the $(l, 1)$-th entry $W_{l1} := |c_l| + \sum_{i=2}^{l} |c_{il}| + \sum_{j=l+1}^{d} |c_{lj}|$; in this way, $W_{l1} = 0$ implies that $x_l$ has no effect on $x_1$. The least squares term, corresponding to variable $x_1$, in the loss function will become $\sum_{k=1}^{m} \left( x_1^k - \mathbf{c}_1^T \mathbf{x}_1^k \right)^2$ where $m$ is the total number of samples. In summary, we have the following optimization problem

$$\min_{\mathbf{c}_1, \mathbf{c}_2, \ldots, \mathbf{c}_d} \quad \sum_{i=1}^{d} \sum_{k=1}^{m} \left( x_i^k - \mathbf{c}_i^T \mathbf{x}_i^k \right)^2$$

$$\text{subject to} \quad \text{trace} \left( e^{W \circ W} \right) - d = 0,$$

where $\circ$ denotes the element-wise product and the constraint enforces acyclicity w.r.t. a weighted adjacency matrix (Zheng et al., 2018). The problem has $(d-1)d^2/2$ parameters to optimize, while the original NOTEARS optimizes $d^2$ parameters. We solve this problem with augmented Lagrangian method where at each iteration the augmented Lagrangian is approximately minimized by Adam (Kingma and Ba, 2014) with Tensorflow. The Lagrange multiplier and the penalty parameter are updated in the same fashion as in the original NOTEARS.

