# OpenReview forum: "Causal Discovery with Reinforcement Learning"
_ICLR.cc/2020/Conference — Accept (Talk)_

### Official Review · AnonReviewer1 · 2019-10-23
**Official Blind Review #1**

**Rating:** 8

**Review:**

This work addresses the task of causal discovery. The proposed contribution is to apply prior work which uses reinforcement learning for combinatorial optimization to structure learning. Specifically, the proposed optimization problem seeks to maximize a penalized score criterion subject to the acyclicity constraint proposed by Zheng, et al. Empirical results show the proposed method performing favorably in contrast to prior art.

Overall I think this is a sensible idea, and the authors do a nice job of exposition, and empirical evaluation.

My concerns are as follows:

* The novelty is somewhat limited, since the paper is combining two previously proposed ideas (combinatorial search and the acyclicity constraint) for structure learning.

* The paper is loose with technical points. Specifically, the authors claim to use the additive noise model, but then make no restrictions on f(). In this setting, it is fairly well known that we can only hope to learn up to the Markov equivalence class (not the fully directed graph), but there is no mention of this in the paper.

With all of this said, I think overall the paper is an interesting addition to the causal discovery literature.

**Experience Assessment:**

I have published one or two papers in this area.

**Review Assessment: Checking Correctness Of Derivations And Theory:**

I assessed the sensibility of the derivations and theory.

**Review Assessment: Checking Correctness Of Experiments:**

I carefully checked the experiments.

**Review Assessment: Thoroughness In Paper Reading:**

I read the paper thoroughly.

---

> ### Author Response · Authors · 2019-11-08
> **We are grateful to the reviewer's effort and the positive comment**
>
> We are grateful to the reviewer's effort and the positive comment on our paper. We are revising the paper by taking into accounts all the reviewers' comments/suggestions, and the revised version will be uploaded at a later time within this week.
>
> * Regarding 'The novelty is somewhat limited, since the paper is combining two previously proposed ideas (combinatorial search and the acyclicity constraint) for structure learning':
>
> These two ideas are indeed important to our RL based approach to causal discovery. Here we would like to briefly discuss the necessity of the acyclicity constraint from Zheng et al. With the proposed penalty weights in our work, Zheng et al.’s acyclicity constraint $h(A)$ is used to guide the RL agent to generate directed graphs ‘closer’ to be acyclic and the indicator function w.r.t. acyclicity aims to induce exact DAGs. The major benefit of $h(A)$, or more precisely, $h(W\circ W)$ ($W$ denotes the weighted adjacent matrix if it exists, e.g., for linear models, and $\circ$ denotes Hadamard product), is its smoothness that enables continuous optimization for structure learning. This property is not utilized in our approach, and we believe other acyclicity functions, which measure certain ‘distance’ of a directed graph to be acyclic and do not need to be differentiable, can also be used here. We will add more discussions on this point in the revision.
>
> * Regarding 'The paper is loose with technical points. Specifically, the authors claim to use the additive noise model, but then make no restrictions on f(). In this setting, it is fairly well known that we can only hope to learn up to the Markov equivalence class (not the fully directed graph), but there is no mention of this in the paper':
>
> Thanks for this helpful comment. We will add a sentence in Section 2 to state this result, along with the fact that we use fully identifiable models to generate observations in our experiments.
>
> We once again appreciate the reviewer’s effort on reviewing our paper.

---

### Official Review · AnonReviewer3 · 2019-10-23
**Official Blind Review #3**

**Rating:** 8

**Review:**

Update: after the revision, I have decided to increase my score to 8.

Original comments:

In this paper, the authors proposed a new reinforcement learning based algorithm to learn causal graphical models. Simulations on real and synthetic data also shows promise.

Pros

1. It's great to see the authors has done a comprehensive comparison with the other methods, especially under different simulation scenarios.

2. The novel idea of applying reinforcement learning to DAG search sounds intriguing. Reinforcement learning offers a powerful tool for policy evaluation and decision making. It’s good to see that the author can successfully extend such toolbox to the field of causal structure learning. To the best of the author’s knowledge, such idea has never been considered by previous work in causal graphical models.

Cons.

1. In the introduction section, the authors claimed that “GES is not guaranteed in the finite sample regime”. This seems to be incorrect. For example, the Nandy et al. paper tackles exactly the finite sample problem.

In conclusion, overall this is a sensible idea, although some of the preliminaries still remain to be polished.

**Experience Assessment:**

I have published one or two papers in this area.

**Review Assessment: Checking Correctness Of Derivations And Theory:**

N/A

**Review Assessment: Checking Correctness Of Experiments:**

I assessed the sensibility of the experiments.

**Review Assessment: Thoroughness In Paper Reading:**

I read the paper at least twice and used my best judgement in assessing the paper.

---

> ### Author Response · Authors · 2019-11-08
> **We thank the reviewer for the positive feedback and would like the reviewer to provide more details**
>
> We thank the reviewer for the positive feedback on our work.
>
> Regarding 'GES is not guaranteed in the finite sample regime' and Nandy et al. paper 'High-dimensional consistency in score-based and hybrid structure learning':
>
> Here we aimed to state the consistency result of GES established by Chickering, and we find that Nandy et al. paper is also about consistency of GES but w.r.t. high dimension settings. In our understanding, consistency means that the probability of correct estimation of the ground truth goes to one as the number of samples approaches infinity, and we believe that this is also the case with Nandy et al. paper (please find below some quoted sentences where $n$ denotes the number of samples).  We however do not find a result or claim regarding guaranteed performance in the finite sample regime. In case we may misunderstand or miss certain results, can the reviewer please give more details on 'the Nandy et al. paper tackles exactly the finite sample problem', and if possible, the corresponding theorems or claims in Nandy et al. paper and other papers as well?
>
> Again we greatly appreciate the reviewer's effort. We would definitely revise our statement if we misunderstand/omit the result of GES in the finite sample regime.
>
> -------------
>
> Quoted sentences from the arxiv version of Nandy et al. paper, available at https://arxiv.org/pdf/1507.02608.pdf:
>
> Page 3: 'In this paper, we prove high-dimensional consistency of GES, and we propose new hybrid algorithms based on GES that are consistent in several sparse high-dimensional settings and scale well to large sparse graphs. To the best of our knowledge, these are the first results on high-dimensional consistency of score-based and hybrid methods.'
>
> Page 7: 'Consistency of $\mathcal S$ assures that $\mathcal G_0$ has a lower score than any DAG that is not in the Markov equivalence class of $\mathcal G_0$, with probability approaching one as $n\to\infty$ (Proposition 8 of Chickering [2002b]).'
>
> Page 19, Theorem 5.2: 'Assume (A1)-(A6). Let $\hat{\mathcal C}_n$, $\breve{\mathcal C}_n$ and $\tilde {\mathcal C}_n$ be the outputs of ARGES-CIG based on $\hat{\mathcal I}_n$, ARGES-skeleton based on $\hat{\mathcal U}_n$ and GES respectively, with the scoring criterion $\mathcal S_{\lambda_n}$. Then there exists a sequence $\lambda_n\to 0$ such that $\lim_{n\to\infty}\mathbb P(\hat{\mathcal C}_n=\mathcal C_{n0})=\lim_{n\to\infty}\mathbb P(\breve{\mathcal C}_n=\mathcal C_{n0})=\lim_{n\to\infty}\mathbb P(\tilde{\mathcal C}_n=\mathcal C_{n0})=1.$'

---

### Official Review · AnonReviewer2 · 2019-10-25
**Official Blind Review #2**

**Rating:** 8

**Review:**

In this paper, the authors propose an RL-based structure searching method for causal discovery. The authors reformulate the score-based causal discovery problem into an RL-format, which includes the reward function re-design, hyper-parameter choose, and graph generation. To my knowledge, it’s the first time that the RL algorithm is applied to causal discovery area for structure searching.

The authors’ contributions are:
(1) re-design the reword function which concludes the traditional score function and the acyclic constraint

(2) Theoretically prove that the maximizing the reward function is equivalent to maximizing the original score function under some choices of the hyper-parameters.

(3) Apply the reinforce gradient estimator to search the parameters related to adjacency matrix generation.

(4) In the experiment, the authors conduct experiment on datasets which includes both linear/non-linear model with Gaussian/Non-gaussian noise.

(5) The authors public their code for reproducibility.

Overall, the idea of this paper is novel, and the experiment is comprehensive. I have the following concerns.

(1) In page 4 Encoder paragraph, the authors mention that the self-attention scheme is capable of finding the causal relationships. Why? In my opinion, the attention scheme only reflects the correlation relationship. The authors should give more clarifications to convince me about their beliefs.

(2) The authors first introduce the h(A) constraint in eqn. (4), and mentioned that only have that constraint would result in a large penalty weight. To solve this, the authors introduce the indicator function constraint. What if we only use the indicator function constraint? In this case, the equivalence is still satisfied, so I am confused about the motivation of imposing the h(A) constraint.

(3) In the last paragraph of page 5, why the authors adjust the predefined scores to a certain range?

(4) Whether the acyclic can be guaranteed after minimizing the negative reward function (the eqn.(6))? I.e., After the training process, whether the graph with the best reward can be theoretically guaranteed to be acyclic?

(5) In section 5.3, the authors mention that the generated graph may contain spurious edges? Whether the edges that in the cyclic are spurious? Whether the last pruning step contains pruning the cyclic path?


(6) In the experiment, the authors adopt three metrics. For better comparison, the author should clarify that: the smaller the FDR/SHD is, the better the performance, and the larger the TPR is, the better the performance.

(7) From the experimental results, the proposed method seems more superiors under the non-linear model case. Why? Could the authors give a few sentences about the guidance of the model selection in the real-world? i.e., when to select the proposed RL-based method? And under which case to choose RL-BIC, and which case to selection RL-BIC2?

(8) What’s training time, and how many samples are needed in the training process?


Minor:
1. In the page 4 decoder section, the notation of enc_i and enc_j is not clarified.

2. On page 5, the \Delta_1 and \Delta_2 are not explained.

3. For better reading experience, in table 1,2,3,4, the authors should bold value that has the best performance.



**Experience Assessment:**

I have published one or two papers in this area.

**Review Assessment: Checking Correctness Of Derivations And Theory:**

I carefully checked the derivations and theory.

**Review Assessment: Checking Correctness Of Experiments:**

I carefully checked the experiments.

**Review Assessment: Thoroughness In Paper Reading:**

I read the paper thoroughly.

---

> ### Author Response · Authors · 2019-11-08
> **We greatly appreciate the reviewer's comments/suggestions [Author Response 3/3]**
>
> (8) 'What’s training time, and how many samples are needed in the training process?'
>
> We did not include the training time because we used different machines for our experiments. The implementation of benchmark methods can also be optimized to reduce time (e.g., DAG-GNN's codes did not work with GPU) and the results may be somehow inaccurate. Here we just provide a rough description with 12-node linear data models:
>
> - Traditional methods PC and GES were run on a laptop with Intel 4-core i7 CPU, and produced the estimated result within 10 seconds;
> - NOTEARS and ICA-LiNGAM were also run on the laptop and can be finished in 1~3 minutes (we set the maximum number of iterations of the ICA algorithm to be 20,000, ten times of the default number used by the ICA-LiNGAM authors);
> - CAM was run on the same laptop and typically required 7~8 minutes;
> - Our algorithms RL-BIC and RL-BIC2 were run with Intel Xeon 3.20GHz CPU and Nvidia Quadro RTX 5000 GPU. Both methods took about 30~40 minutes with 12-nodes graphs and 20,000 iterations. For 30-node graphs and 30,000 iterations, they needed around 3 hours;
> - DAG-GNN took about 1 hour with the same Intel Xeon 3.20GHz CPU (their codes with GPU option did not work; the algorithm in fact did not require such a long time to reach convergence, yet no early stopping choice was provided in the codes);
> - GraN-DAG with the same CPU and GPU took about 20~30 minutes.
>
> Regarding the sample number, we have given the number of samples in each experiment description.
>
> (9) Minor:
>
> 1. 'In the page 4 decoder section, the notation of enc_i and enc_j is not clarified. '
>
> Actually $enc_i$ is given in the last sentence of the encoder part.
>
> 2. 'On page 5, the \Delta_1 and \Delta_2 are not explained.'
>
> Thanks for pointing this out. We will add a definition for the two notations.
>
> 3. 'For better reading experience, in table 1,2,3,4, the authors should bold value that has the best performance.'
>
> Thanks for this suggestion. We have considered doing so, but it is usually the case that a method that has the best TPR does not achieve the lowest FDR, and only making one in bold seems insufficient to evaluate the overall performance of a method. If possible, can the reviewer give further suggestion on this part? Thanks.

---

> ### Author Response · Authors · 2019-11-08
> **We greatly appreciate the reviewer's comments/suggestions  [Author Response 2/3]**
>
> (3) ‘In the last paragraph of page 5, why the authors adjust the predefined scores to a certain range?’
>
> We observe that the acyclicity penalty terms do not depend on the particular score functions. Consequently, even we have started with small penalty weights and then gradually increase them, the initial penalty weights may still be too high for other scores on the same problem, e.g., the independence based score function, which in the ideal case shall be zero, or if one wants to use the sample average BIC score. Therefore, we adjust the score to a certain range so that the RL algorithm with some choice of penalty weights is likely to work for other score functions as well.
>
> (4) 'Whether the acyclic can be guaranteed after minimizing the negative reward function (the eqn.(6))? I.e., After the training process, whether the graph with the best reward can be theoretically guaranteed to be acyclic? '
>
> This is also a good point. In theory, no, since policy gradient methods only guarantee local convergence. But with the proposed strategy for penalty weights, the inferred graphs from RL algorithms are all DAGs in our experiments. In practice, if the graph from the training process is not acyclic, we may rerun the algorithm, possibly add more penalty weights, and/or try different NN weights as well. Post-processing method like pruning can also be used to make the inferred graph acyclic.
>
> (5) 'In section 5.3, the authors mention that the generated graph may contain spurious edges? Whether the edges that in the cyclic are spurious? Whether the last pruning step contains pruning the cyclic path? '
>
> We do not fully understand this question, but we try to address it as much as we can.
>
> Spurious edge means false discovery, i.e., an edge in the estimated graph does not exist in the true graph. Using BIC, negative log likelihood, or other reconstruction error based score functions is very likely to result in spurious edge in finite sample regime. For example, the least squares loss would not increase, and usually decreases, if we include a non-parental node when fitting a causal relation. If this additional edge caused by this node does not violate the acyclicity constraint, we indeed get a better reward. So in practice with finite samples, spurious edges are hardly avoided and post-processing is needed.
>
> We consider majority vote to somehow remove spurious edges based on the observation that the top few graphs, ranked by their rewards, are usually structurally similar. However, a majority vote of several DAGs is not necessarily a DAG. As per our assumption that the true graph is a DAG, a cyclic path must contain at least one spurious edge, but spurious edge does not necessarily lie in a cyclic path. Since the pruning methods with a decreasing tolerance or an increasing threshold can lead to the empty graph, i.e., graphs that have no edges, we believe that, with proper tolerance or threshold, the methods will result in DAGs so that the cyclic path is removed.
>
> Please let us know if we have addressed your concern.
>
> (6) 'In the experiment, the authors adopt three metrics. For better comparison, the author should clarify that: the smaller the FDR/SHD is, the better the performance, and the larger the TPR is, the better the performance.'
>
> Thanks. We will add this clarification in the revised version.
>
> (7) 'From the experimental results, the proposed method seems more superiors under the non-linear model case. Why? Could the authors give a few sentences about the guidance of the model selection in the real-world? i.e., when to select the proposed RL-based method? And under which case to choose RL-BIC, and which case to selection RL-BIC2?'
>
> This is an acute observation. We did not notice it previously. We do not think that this should be the case, although it appears so. Different methods may have different assumptions on data generating procedures, and if the ground truth meets the assumptions, these methods usually perform very well (but may still incur estimation errors due to finite samples). For example, ICA-LiNGAM recovers all the true edges without any false discoveries for LiNGAM data. For non-linear model cases, it may be because we use Gaussian process regression which is nonparametric and can fit causal relations well.
>
> As to the model selection, we believe that this is related to what score functions perform well here. For example, if we know that the true data model does not follow an additive model, then it is very likely that the least squares loss or BIC is not appropriate. For RL-BIC or RL-BIC2, model selection then reduces to whether we shall use the least squares or negative log likelihood as our loss function.

---

> ### Author Response · Authors · 2019-11-08
> **We greatly appreciate the reviewer's comments/suggestions [Author Response 1/3]**
>
> We greatly appreciate the reviewer's comments/suggestions, many of which will lead to a more readable and self-contained version of our paper. We attempt to address all the concerns in the following. In case we may omit certain places, please do let us know. The revised manuscript will be uploaded at a later time within this week.
>
> (1) 'In page 4 Encoder paragraph, the authors mention that the self-attention scheme is capable of finding the causal relationships. Why? In my opinion, the attention scheme only reflects the correlation relationship. The authors should give more clarifications to convince me about their beliefs.'
>
> This is a good point. The statement in the submitted manuscript is indeed vague and confusing. We agree with the reviewer that the attention scheme reflects only correlation or association. Many existing score based methods exploit correlations together with structural constraints to discover the causal relations. For example, NOTEARS uses linear regression for fitting the causal function with least squares as loss function. Clearly, using only linear regression could not find causal relationships, and what enables linear regression to find causal graphs is the acyclicity constraint. Since the self-attention scheme is very powerful in capturing the (correlated or associated) relations amongst variables, we believe that it, together with the acyclic constraint, is capable of finding causal relationships. We will revise the statement accordingly. Thanks very much for this comment that makes our paper more rigorous.
>
> (2) ‘The authors first introduce the h(A) constraint in eqn. (4), and mentioned that only have that constraint would result in a large penalty weight. To solve this, the authors introduce the indicator function constraint. What if we only use the indicator function constraint? In this case, the equivalence is still satisfied, so I am confused about the motivation of imposing the h(A) constraint.’
>
> This is very insightful. With only the indicator function term, problems (1) and (6) can still be equivalent. Yet this fact does not imply that an RL algorithm would also work well. Actually our initial reward consisted of the score function and only the indicator term, which worked well for small graphs (with $\leq 6$ nodes or so) but very poorly for larger ones. We observed that the RL algorithm, with randomly initialized NN weights, could hardly generate DAGs in this case when only the indicator term was used. We now attempt to illustrate why this is the case:
>
> (a) the directed graphs in our approach are randomly generated according to Bernoulli distributions, and without loss of generality, consider that each edge is drawn independently according to Bern(0.5). For small graphs (with $\leq 6$ nodes), a few hundreds of samples of directed graphs are very likely to contain a DAG. Yet for large graphs, the probability of sampling a DAG is a lot lower. If no DAG is generated during training, then the RL agent can hardly learn to generate DAGs. Thus, we need the reward to guide the agent to produce DAGs. This, however, is difficult for large graphs with only the indicator term; see below.
>
> (b) for a cyclic directed graph with all possible directed edges in place and a cyclic directed graph with only two edges (that is, $i\to j$ and $j\to i$ for some $i\neq j$), the latter is 'closer' to be acyclic in some sense, e.g., number of edge operations to make it acyclic. However, the first one is likely to have a lower BIC score when using linear regression for fitting causal relations, and yet the penalty terms of acyclicity are the same. In other words, the first graph usually has a better reward, which does not help the agent to tend to generate DAGs. This fact motivates us to include the other penalty term that measures some 'distance' to be a DAG, so that the agent can be trained to produce graphs closer to acyclicity and finally to generate exact DAGs. With initialized NN weights, the generated graphs at early iterations can be 'far' from acyclicity for large problems, and we believe that using only the indicator function is insufficient.
>
> A question is then what if we start with a DAG, e.g., by initializing the probability of generating each edge to be very small. This setting did not lead to good performance, either. The generated directed graphs at early iterations can be very different from the true graphs in missing many true edges, and the resulting score is much higher than the optimum under the DAG constraint. With small penalty weights of the acyclicity terms, the agent would produce cyclic graphs with lower scores, which then reduces to case (b). On the other hand, large penalty weights, as we have discussed in the paper, limit exploration of the RL agent and usually result in DAGs whose scores are far from optimum.
>
> We hope that the above discussion has addressed the reviewer’s concern. We will add more discussions to make this point clear in the revised paper.

---

### Author Response · Authors · 2019-11-11
**We have uploaded a revised version**

Dear reviewers,

We have uploaded a revised version of our paper, following the suggestions/comments from all the reviewers. Some changes are:

- Section 3, Page 3: we add a sentence on the identifiability of Markov equivalence class, following Reviewer 1's comment;
- Section 4, Page 4: we revise the statement on self-attention scheme being capable of finding causal relationships, following Reviewer 2's suggestion;
- Section 5, Page 5: we add a sentence on the necessity of the acyclicity constraint from Zheng et al., according to Reviewer 1's comment;
- Section 5, Page 5: we add a sentence on the effect of picking $\lambda_2=0$ or using only the indicator function in the reward, with a more detailed discussion given in Appendix C in the revised manuscript, according to Reviewer 2's comment;
- Section 5, Page 5: we add definitions for $\Delta_1$ and $\Delta_2$ and rephrase the paragraph for a better presentation, following Reviewer 2's comment;
- Section 6, Page 6: we add a sentence to state that a lower SHD indicates a better performance, according to Reviewer 2's comment.

To Reviewer 3, we have not revised our statement on the finite sample result of GES in the new manuscript. We are eager to learn this finite sample result of GES and are happy to modify the statement if we misunderstand or omit related results in Nandy et al. paper.

We once again thank all the reviewers for their effort and many helpful comments/suggestions.

---

### Author Response · Authors · 2019-12-27
**About Codes and Datasets**

[update: 03/19/2020]

We have released our codes, along with datasets and training logs in the paper, at https://github.com/huawei-noah/trustworthyAI/tree/master/Causal_Structure_Learning/Causal_Discovery_RL .

Please file an issue if you have any questions.

---------------

Hi all,

Our codes and datasets are currently undergoing the regular open-source process of Huawei Noah's Ark Lab, and will be made available as a repository at https://github.com/huawei-noah. We will also release the training logs of the experimental results that are reported in the paper.

We will let you know once the codes are released.

Best Regards,
Shengyu

---

### Public Comment · ~Le_Song1 · 2020-02-02
**Related work**

This is a very interesting paper on using reinforcement learning to learn to solve combinatorial optimization problems involving graphs, in this particular case, the structure of the graphical models.

There are two highly relevant papers which are worthwhile discussing in context and can enrich the current paper:

1. Learning Combinatorial Optimization Algorithms over Graphs. Hanjun Dai, Elias B. Khalil, Yuyu Zhang, Bistra Dilkina, Le Song. NeurIPS 2017.

2. GLAD: Learning Sparse Graph Recovery. Harsh Shrivastava, Xinshi Chen, Binghong Chen, Guanghui Lan, Srinivas Aluru, Han Liu, Le Song. ICLR 2020.

---

> ### Author Response · Authors · 2020-02-03
> **Re: Related work**
>
> Thanks for letting us know. My first impression is that the first one is indeed very relevant. We will have a careful read of both papers.

---

### Decision · Program_Chairs · 2019-12-19

**Decision:**

Accept (Talk)

**Comment:**

This paper proposes an RL-based structure search method for causal discovery. The reviewers and AC think that the idea of applying reinforcement learning to causal structure discovery is novel and intriguing. While there were initially some concerns regarding presentation of the results, these have been taken care of during the discussion period. The reviewers agree that this is a very good submission, which merits acceptance to ICLR-2020.